# Monetising Resources on a SoLiD Pod Using Blockchain Transactions

Hendrik Becker[0000−0003−4987−481X], Hung Vu[0000−0003−1983−8797], Anett Katzenbach[0000−0003−1895−8316], Christoph H.-J. Braun[0000−0002−5843−0316], and Tobias Käfer[0000−0003−0576−7457]

Institute AIFB, Karlsruhe Institute of Technology (KIT), Karlsruhe, Germany
{ukuiq,ublxf,ususm,uvdsl}@student.kit.edu
tobias.kaefer@kit.edu

**Abstract.** Our demo showcases a system that allows users to provide access to web resources in exchange for payments via the blockchain. The system enables users to create offers for their resources or buy access rights for resources belonging to other users. Access rights can be granted only for a limited amount of time. We built our system as SoLiD Pods and Apps: We developed two server modules for SoLiD Pods that automatically (1) grant access for valid payments via the blockchain and (2) remove expired access rights. On top, we developed a SoLiD App that allows to offer resources, browse and request offered resources, and make payments via the blockchain.

**Video** http://people.aifb.kit.edu/co1683/2021/eswc-demo-solibra/#v
**Code** https://github.com/bright-fox/SolidBlockchain

## 1 Introduction

In today's web, companies keep user data in centralised platforms, outside of the control of the user. Users are thus unable to control, use and profit from their own data. Multiple recent projects want to tackle this issue: First, Jack Dorsey, the CEO of Twitter, a microblogging platform, recently announced Twitter's effort of re-decentralisation based on decentralised data and open standards for the benefit of all Twitter users[1]. In addition, he mentions blockchain technology as one potential building block, especially regarding monetisation.

Second, Tim Berners-Lee's on-going endeavour to re-decentralise the web manifests in the Social Linked Data project (SoLiD). The SoLiD specification[2] provides the basis for decoupling data and applications, while ensuring data privacy through user-defined access control. Users store their data in personal online data storages (Pods) and define access control on their resources as they wish.

---

[1] https://twitter.com/jack/status/1204766078468911106
[2] https://solid.github.io/specification/

Also in the decentralised spirit, blockchain technologies, as pioneered by Bitcoin [5], allow for peer-to-peer transactions of money. In the context of those decentralised technologies, we built a proof-of-concept that is set in a decentralised web scenario and allows for trading access to web resources via the blockchain.

Our approach is not limited to a specific use-case or specific types of resources: As long as a resource can be stored on a SoLiD Pod, users can monetise it. Users decide, if and under which conditions they are willing to share their resources by specifying a price and a duration for which access will be granted.

In this demo, we showcase:
- A general approach to enable users to monetise digital resources stored on SoLiD Pods using blockchain transactions.
- An implementation[3] that automates access control management based on payment and time.

We built our demonstration as SoLiD Apps that communicate with SoLiD Pods and the Ethereum network, a blockchain implementation. The server part of our implementation builds on SoLiD's semantic access control (ACL) rules to automatically update resource access based on payments and elapsed time. Resource access is granted upon incoming payments and revoked when access rights expire. We use Resource Description Framework (RDF) to model resource offers, purchases and user notifications.

This paper is structured as follows: First, we give a short overview on related work. Next, we present the system's architecture and cover briefly data modeling. Then, we showcase a walkthrough for the demonstrator. Finally, we conclude.

## 2    Related work

We work in the intersection of SoLiD, blockchain, and monetisation on the web. Seminal writings in each area include an early description about the SoLiD project [4], foundational articles about Bitcoin [5] and Ethereum [2] as two blockchain implementations that allow for peer-to-peer transactions, and the recent Patreon platform[4] for selling and buying digital artistic work on the web.

The intersection is laden with visions, and less with solutions: [8] presents the vision of an app for musicians to sell their songs to others. The app should use SoLiD for data management, and a blockchain facilitate payments. Our demo is an implementation of a more general version of this idea. [3] presents the vision of a data marketplace to fund the maintenance of the infrastructure for decentrally provided Linked Data. Our demo is not about a marketplace as central intermediary but instead about using blockchain for decentralised payment. We found descriptions of solutions only where the solution is in the intersection blockchain+SoLiD, without the monetisation aspect. Consider for instance [6] who examine different approaches in SoLiD-based applications using blockchains and focus on the combination of the two technologies to make data

---

[3] A link to the code can be found on page 1.
[4] https://www.patreon.com/

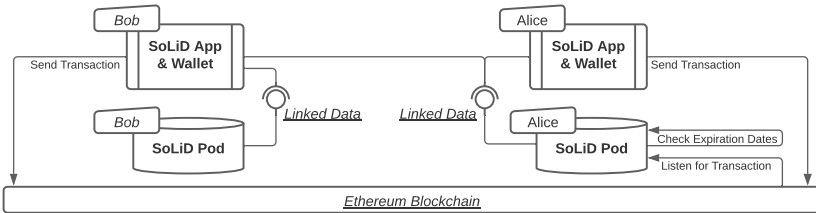

**Fig. 1.** The system architecture. One SoLiD Pod and wallet per user.

verifiable through transparency without losing data sovereignty. Similarly, [1] uses Linked Data and the blockchain for verification of data along a supply chain.

Our approach uses Web Access Control (WAC), the current default technology from SoLiD, as basis to provide simple access control. There are more sophisticated approaches, which, e. g., also allow for usage control, such as the Open Digital Rights Language (ODRL)[5]. In previous work, ODRL has been proposed as basis for a Policy-Driven Linked Data Market, where Linked Data can be offered and requested, and agreements can be described and enforced [7].

## 3   System Architecture

Figure 1 illustrates the components of this demo. The demo set-up consists in a SoLiD App with two UIs for buyer and seller of a resource. Those UIs both consume SoLiD Pods and have access to the Ethereum blockchain to send transactions. The SoLiD Pod of the buyer maintains the WebID and wallet address of the buyer and a list of pending purchasing requests. The SoLiD Pod of the seller maintains the WebID and wallet address of the seller, the resources offered next to their descriptions and ACLs. This Pod has specialised modules to (1) access the blockchain to listen for transactions and (2) modify the ACLs according to time and the transactions.

## 4   Data Modeling

We model the data in RDF using terms from a range of vocabularies: We use (1) schema.org[6] to model offers and pending resource purchases including price and corresponding currency; (2) OWL Time[7] to describe the duration of the access to resources; (3) Linked Data Notifications[8] (LDN) for notifications about

---

[5] https://www.w3.org/TR/odrl-model/

[6] https://schema.org/

[7] https://www.w3.org/TR/owl-time/

[8] https://www.w3.org/TR/ldn/

incoming payments; (4) EthOn[9] to describe (a) blockchain transactions such that a resource owner can verify the validity of the purchase request, and (b) to link the address of the cryptocurrency wallet to the user's SoLiD profile; (5) WAC[10] to define ACL rules including read, write, and append rights for specific agents or agent groups, which we augment with OWL Time for durations.

## 5    Walkthrough for the Demonstrator

Our demo shows how our system can be used by Alice who offers a resource, and by Bob who buys time-restricted access to it, cf. the video linked on Page 1.

**Resource and offer creation** Using the UI, Alice uploads a resource to be sold to her SoLiD Pod and defines the resource's price and access duration. The corresponding offer with the defined conditions is automatically created on her SoLid Pod. For now, only Alice has access rights for this resource.

**Offer retrieval** Bob uses his UI to browse all available offers for resources from Alice on her SoLiD Pod. The UI filters out resources, for which Bob has a pending request or which Bob has already bought.

**Resource purchase** Bob selects an offer and sends a transaction to Alice's wallet via the blockchain with the specified price. The transaction also contains the offered resource's URI, Bob's WebID, the price and duration. This enables Alice to verify Bob's blockchain transaction for the purchase. The UI automatically sends an LDN to Alice to inform her of the purchase and stores a pending resource purchase in Bob's SoLiD Pod for record keeping.

**Access grant** Alice's Pod runs a server module, which continuously monitors the LDNs in her inbox for access requests for her offers. The LDNs are compared to the information of offers in the Pod and transactions on the blockchain. If the information from the offer, the transaction and the LDN matches, the access rights of Bob are added to the ACL of the resource. Approving of access requests can also be done by Alice manually through the access control panel of the user interface.

**Access rights housekeeping** Alice's Pod runs another server module that continuously checks for and removes expired access rights.

## 6    Conclusion

In this demo, we presented a general approach to enable monetisation of web resources stored on SoLiD Pods using blockchain transactions. Our implementation automatically grants access rights upon incoming payments and revokes expired access rights. Of course, SoLiD Pod extensions like our ACL-updating server module need to be trusted by the trading parties. In particular, the party running the Pod software needs to be trusted, who would be responsible for

---

[9] https://ethon.consensys.net/
[10] https://www.w3.org/wiki/WebAccessControl

deploying the module. In a completely decentralised web setting, this trust is not trivial to establish. Just as there is no certification process to go through before a web server implementation may be connected to the internet, there is no certification process for SoLiD Pod software. Even if the code had been certified, it needs to be made sure that the code eventually executed is the same as the certified code. Code signing has been proposed to deal with such issues. In the physical world, governments require organisations in the financial business (e. g. banks) to go through a vetting process before they are allowed to offer the services and take part in the federated financial services system. Similarly, in the virtual world, if your web server uses some (centralised/federated) payment service (e. g. MasterCard) on the web, you need to show to the respective financial service provider that you adhere to security and privacy guidelines. Also, if you commercially trade on the web in Europe you need to follow the national implementation of the EU directive 2000/31/EG about e-commerce, which for instance in some countries requires you to state your name and address such that, e. g. customers can lay charges against you, and this possibility introduces some trust. For a SoLiD Pod hosting provider this could mean that users need to go through a know-your-customer process before users may use the services.

**Acknowledgements**

This work is partially supported by the German federal ministry of education and research (BMBF) in TraPS, a Software Campus project (FKZ 01IS17042).

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
