# OpenReview forum: "Monetising Resources on a SoLiD Pod Using Blockchain Transactions"
_eswc-conferences.org/ESWC/2021/Conference/Poster_and_Demo_Track — ESWC2021 P&D_

### Official Review · AnonReviewer2 · 2021-04-14
**Nice demonstration of the use of SW in combination with other technologies to create an useful application**

**Rating:** 7
**Confidence:** 4

**Review:**

The paper describes a demo system that combines different technologies (including SW) to enable a decentralized, monetized and privacy-preserving application to sell/buy resources.

It makes a compelling argument of the need of a decentralized solution, referring to works in Twitter and the SoLiD project. The demo uses a combination of SoLiD Pods, Blockchain (Ethereum) and RDF models, to describe, sell and buy resources, as well as to perform payments and set time limit on the use of resources.

The paper is accompanied by a video of the demo that will be shown, if accepted.

The paper clearly describes the proof of concept, and I believe the demo would be relevant to a large audience to the conference.



**Anonymity:**

Yes, I would like my review to remain anonymous.

---

### Official Review · ~Sabrina_Kirrane2 · 2021-04-14
**Timely demo proposal using up and coming technologies**

**Rating:** 8
**Confidence:** 5

**Review:**

This paper demonstrates how the SoLiD platform together with an Ethereum Blockchain application can be used to monetise resources.

The paper is well written, the supporting video is easy to follow, and the proposed demo fits very nicely with the conference.

I really like the work, however considering the access control policies are a major part of the proposal I was disappointed that the related work section didn’t provide the relevant background.

For instance, the following paper demonstrates how ODRL can be used not only to represent access policies but also to specify access requests, offers and agreements, and propose an approach to generate on-the-fly contracts that govern all explicit and implicit non-enforceable policies:

Steyskal, S. and Kirrane, S., 2015, September. If you can't enforce it, contract it: Enforceability in Policy-Driven (Linked) Data Markets. In Semantics (posters & demos).

Also, the existing proposal talks about simple access control, however I would argue that what is needed here is usage control in the general sense, which is a broader team encompassing access constraints, licenses, privacy preferences, etc... Once I have given access I wonder if it is actually possible to take away access, considering that the buyer can download that resource thus decoupling it from the access control enforcement platform (see the discussion in the above paper).

Although the following paper focuses on consent, transparency, and automated compliance checking, and thus is further away from the existing proposal, it may be interesting from a future work perspective:
Havur, G., Vander Sande, M. and Kirrane, S., 2020. Greater Control and Transparency in Personal Data Processing. In International Conference on Information Systems Security and Privacy (ICISSP).

Minor comments:
- “Our demo showcases a system using which users can provide access to web resources to other users in exchange for a payment via the blockchain.” -> this sentence needs to be rephrased.

- “buy access rights for resources of other users.” -> buy access rights to resources, belonging to other users? buy access rights to resources, published by other users? Other?

- “As long as a good can be stored” -> As long as a resource can be stored.

- RDF should be expanded on first usage.

- “An implementation that automates access control management based on payment and time” -> the full stop at the end of the sentence is missing.

**Anonymity:**

No, I would like my review to be deanonymized.

---

### Official Review · AnonReviewer3 · 2021-04-16
**This demo shows an implementation  of the system which monetizes resources Resources on a SoLiD Pod Using Blockchain Transactions**

**Rating:** 6
**Confidence:** 2

**Review:**

This demo shows an implementation  of the system which monetizes resources Resources on a SoLiD Pod Using Blockchain Transactions.
This is a good example to use important techniques for decentralized web which are attracted by many people.
The feature of the proposed system is that the authors mode the data in RDF using  well-known vocabularies such as schema.org, OWL Times.
However, I think it is more valuable if why they use RDF for the modeling with benefits using RDF terms.

The implementation looks good and solid through its system architecture and the demo video, while I'm not so familiar with SoLiD Pod and Blockchain.



**Anonymity:**

Yes, I would like my review to remain anonymous.

---

### Official Review · AnonReviewer1 · 2021-04-16
**A solid and blockchain-based architecture for transaction applications**

**Rating:** 7
**Confidence:** 4

**Review:**

This papers presents an architecture for building solid-based user applications for blockchain-based monetary transactions for resources on the Web.

The architecture describes a clear usage of the Solid capabilities (LDN, ACL, etc) plus an interesting integration with a distributed ledger such as ethereum. I would like to read the authors perspective on the third-party module trust issue they raise in the conclusions section. How could this issue be addressed?

Also is a pity that the source code of the applications is not made public in the paper.



**Anonymity:**

Yes, I would like my review to remain anonymous.

---

### Decision · Program_Chairs · 2021-04-19

Accept